# Social Acceptance in Physical Education and the Regular Classroom: Perceived Motor Competency and Frequency and Type of Sports Participation

**DOI:** 10.3390/children10030568

**Published:** 2023-03-16

**Authors:** Anne G. M. de Bruijn, Femke van der Wilt

**Affiliations:** Faculty of Behavioural and Movement Sciences, Vrije Universiteit Amsterdam, 1081 HV Amsterdam, The Netherlands

**Keywords:** social preference, peer relationships, primary school, physical competence, motor competence, sports participation, physical education

## Abstract

This study examined relations of primary school children’s perceived physical competence and sports participation (frequency and type) with social acceptance in the regular classroom and physical education (PE) and whether these relations differed depending on the type of sport children participated in (team vs. individual sports). In total, 182 children (48.9% boys, mean age 9.90 years, SD = 1.23) filled out questions on their perceived physical competence and sports participation and indicated three peers with whom they liked/disliked working in PE and the regular classroom. Multilevel structural equation models in Mplus showed that frequency of sports participation was positively related to social acceptance in the context of PE. Additionally, for children in team sports, the frequency of sports participation was related to their social acceptance in PE, whereas for children in individual sports, perceived physical competence was related to social acceptance in PE. No relations were found in the regular classroom. Relations of perceived physical competence and sports participation with social acceptance seem to depend on the school context and the type of sport involved. In designing PE classrooms, children’s physical competence and sports participation seem essential factors to take into account to provide all children with positive social experiences.

## 1. Introduction

Engaging in positive peer interactions is crucial for children’s learning and development [1]. One key factor in facilitating such interactions is a child’s social acceptance [2]. As a consequence, peer rejection—by excluding children from positive peer interactions—can have devastating consequences for children’s development, being associated with loneliness and anxiety, aggressive behaviour, academic failure and eventually school dropout [1]. Because of the significance of social acceptance, it is of vital importance to identify its correlates.

Previous studies have mainly focused on (correlates of) social acceptance within the regular primary school classroom. Yet, less structured environments, such as physical education (PE), might provide unique opportunities for gaining social acceptance. That is, the inherently social character of PE supports the development of social and interpersonal skills necessary to build positive peer relationships and, thereby, obtaining acceptance from peers [3,4,5]. Few studies, however, have focused on the context of PE, despite its expected benefits for children’s development of social skills and relations [3,6,7]. Therefore, in the present study, we investigated correlates of social acceptance within the primary school PE context, focusing on perceived physical competence and sports participation. By examining correlates of social acceptance in PE specifically, comparing it to the general classroom context, we aim to provide insights into the unique opportunities that PE might offer in providing all children with positive social experiences during the school day.

### 1.1. Correlates of Social Acceptance in Different School Contexts

Social acceptance indicates the extent to which a child is liked by peers [8,9]. Previous studies focusing on the regular primary school classroom indicated that children’s academic ability is one of the strongest predictors of their social acceptance [1]. This can be explained by the fact that academic ability is required for successful participation during class [10]. However, by examining correlates of social acceptance in the general classroom, these studies have not taken contextual characteristics into account; whereas, school contexts notably differ from one another [11,12]. For example, compared to the regular classroom, PE provides more opportunities for children’s own input, allows for closer peer interactions and requires specific emotional and communication skills [13]. Additionally, the context is crucial in what is considered appropriate social behaviour [14]. Skills considered adaptive for gaining social acceptance in the regular classroom might, thus, differ from those required in PE [15]. This study, therefore, focused on potential correlates of social acceptance in the regular classroom and PE.

### 1.2. Perceived Physical Competence

An important factor to take into account when examining social acceptance in the PE context is children’s perceived physical competence: children’s belief in and awareness of their own capabilities regarding physical tasks and situations [16]. Although actual physical competence is seen as a prerequisite for successful performance in physical activity contexts, thereby, being a determining factor in primary school children’s social acceptance [10], perceived physical competence is thought to be linked to children’s social acceptance as well [17,18,19,20,21]. That is, perceived physical competency is considered a key factor in children’s experiences during physical activity [22] and several studies have suggested that children’s perceived competence mediates the relation between children’s actual physical competence and physical activity behaviour [23,24,25,26,27,28,29], underlining the important role that perceived physical competence plays in children’s (physical) development. Yet, few studies have considered children’s own perception of their physical competency when examining their acceptance amongst peers.

The potential relation between perceived physical competency and social acceptance can be explained by the fact that children who perceive themselves as physically competent behave accordingly. That is, PE behaviours of children and adolescents are determined to a large extent by their own perceived physical competence [30,31,32], even more so than by their actual competence [23,24]. For example, compared to children with lower perceived physical competency, children with high perceived physical competence put in more effort during primary school PE and are more likely to collaborate with peers [23]. As peers tend to base their opinion on the behaviours they perceive, the extent to which children are socially accepted might not only be predicted by their actual physical competence level, but even more so by their own perception of their physical competence [23]. Given the context specificity of social relations, the potential relation between perceived physical competence and social acceptance can be expected to be stronger in PE than in the regular classroom [10].

### 1.3. Sports Participation

A second important factor to consider when examining social acceptance in PE is children’s sports participation. First, in primary school, children highly value sports, with many children naming “being good at sports” as the most important means for gaining social acceptance and building peer relationships [18,33,34,35]. As engaging in activities that are highly valued by peers is a means for becoming socially accepted [36], children who frequently participate in sports are more likely to be accepted by their peers [37,38,39]. Secondly, sports settings provide opportunities for engaging in social activities and developing social skills [40]. Consequently, children and adolescents who frequently participate in sports are generally more socially competent [41], more empathic [42], and show more prosocial behaviours [43,44]; whereas, low levels of physical activity participation have been related to difficulties with peers, amongst others bullying [43]. As social competence is a major predictor of social acceptance [36], children who participate in sports more frequently are likely to be more socially accepted by peers than their peers who are less engaged in sports activities. Lastly, frequently participating in sports contributes to the development of physical competence during childhood and adolescence, which, in turn, is an important predictor of likeability amongst peers [5,18,21,36,45,46,47,48]. Hence, the frequency in which children engage in sports can be expected to be related to social acceptance (i.e., (partly) through physical competence). In agreement with this idea, previous studies indeed indicated a positive relation between the frequency of sports participation and social acceptance in adolescence [38,39]. Yet, studies were mostly conducted in the general school context, not considering whether the importance of sports participation for children’s social acceptance might differ depending on the context involved. Additionally, most studies were conducted during adolescence, not considering that the development of peer relations already starts in the childhood years.

In assessing the relation between sports participation and social acceptance, the type of sports children participate in is of importance. That is, some types of sports are typically considered more popular by children than others [18]. In addition, some types of sports contribute more strongly to the development of social competence than others. Specifically, because of their inherently social nature, team sports provide more opportunities to develop social competence and build social relations than individual sports [49,50,51]. Children who are members of a team, therefore, have more social experiences, feel more included and experience more social support than children who participate in individual sports [50]. Given the link between social competence and social acceptance [52], the type of sports children are engaged in might be related to the extent to which children are accepted by peers (i.e., through their level of social competence). In addition, as team sports provide children with opportunities to develop their skills and relations, more frequent participation in sports may be a stronger predictor of social acceptance than for children participating in individual sports [49,50,51].

### 1.4. This Study

Previous studies have provided suggestions for a relation between perceived physical competence and social acceptance and between sports participation (both frequency and type) and social acceptance in childhood and adolescence. However, evidence is scattered and the relations between perceived physical competence, sports participation and social acceptance of primary school children have not yet been examined simultaneously. Moreover, despite the unique characteristics of primary school PE compared to the general classroom, it is unclear whether the PE context may provide unique opportunities for gaining social acceptance among peers.

The present study, therefore, investigated whether primary school children’s perceived physical competence and sports participation (frequency and type) are related to social acceptance within two settings: the regular classroom and PE. Based on previous findings, we expected that (1) children who perceive themselves as more physically competent have higher levels of social acceptance, (2) children who are more frequently engaged in sports (frequency) and who are engaged in team sports (type) have higher levels of social acceptance and (3) the relations between perceived physical competence and social acceptance and between sports participation and social acceptance are stronger in PE than in the regular classroom. In addition, using exploratory analyses, we examined whether the relations between perceived physical competence, frequency of sports participation and social acceptance differ depending on the type of sports. We expected that the frequency of sports participation would be more strongly related to social acceptance for children engaged in team sports compared to children in individual sports. We had no specific hypotheses regarding perceived physical competence.

## 2. Materials and Methods

### 2.1. Participants

In total, 182 children (48.9% boys, *n* = 89) of grades 3 to 6 of three Dutch primary schools (12 classes) participated in this study. Mean age of the participating children was 9.90 (SD = 1.23) years. For all participating children, informed consent was provided by their parents. The study was approved by the authors’ faculty (details removed for blind review).

### 2.2. Instrumentation

Social Acceptance. Social acceptance was measured with the sociometric method [53]. The following two questions were used for PE: “who do you like to have on your team in PE?” (positive nomination) and “who do you not like to have on your team in PE?” (negative nomination). Additionally, two questions referred to the regular classroom: “with whom do you like to work in the classroom?” (positive nomination) and “with whom do you not like to work in the classroom?” (negative nomination). Children were asked to nominate three peers per question. Positive and negative nominations scores were obtained for each child by counting the received number of positive and negative nominations, respectively. To control for differences in classroom size, nomination scores were standardised within classroom. Social preference scores were calculated by subtracting the standardised positive nomination score from the standardised negative nomination score. The resulting social preference score was again standardised within classroom. The peer nomination procedure is a reliable method for measuring peer relationships (test–retest reliability is 0.79) [54].

Perceived physical competence was measured with a Dutch translation of the perceived motor competence questionnaire for children (PMC-C) [55]. The PMC-C consists of 24 items measuring children’s perceived competence in key aspects of motor skills: locomotor skills (e.g., running) and object-control skills (e.g., throwing and catching). Children are asked to indicate how well they perform these skills using a 4-point Likert scale in which answer options are indicated by smileys (ranging from sad to extremely happy). An example item is “I am good at throwing a ball”. The mean score on the 24 items indicated perceived physical competence. The PMC has shown to be a reliable (α = 0.79–0.91) and valid measure of children’s perceived physical competence [55]. Internal consistency in the present study was high (α = 0.90).

To measure sports participation, a questionnaire developed for the Dutch National Assessment of PE was used [56]. This questionnaire has been designed to examine the amount of leisure time dedicated to sports of Dutch primary school students. In the present study, children were asked to indicate whether they were member of a sports club and, if yes, which sport they participated in (choosing from a list of options, e.g., ball sports; athletics; dance) and how many times a week they went to their sports club (with options ranging from less than 1 day a week to 7 days a week. In case children participated in multiple sports, answers concerning their main sport (i.e., the sport they participated in most days of the week) were included. Test–retest reliability of these questions has shown to be high (Kappa > 0.9) and the questionnaire has proven to be a valid measure of children’s sport frequency [57]. We classified children’s sports participation as team sports or individual sports using the classification system of Chelladurai and Saleh [58], which classifies sports according to the degree to which performance outcomes depend on involvement of other group members [59]. Sports that are characterised by a high level of interdependence were classified as team sports (e.g., ball sports), whereas sports that are performed highly independent (e.g., dance) were classified as individual sports [59].

### 2.3. Procedure

Questionnaires on children’s social acceptance, perceived physical competence and sports participation were conducted in their own classroom by supervised research assistants using a standardised instruction protocol. Children filled out the questionnaires individually in silence. They were allowed to ask for clarification by the researchers in case they did not understand a question. In total, children needed approximately 45 min to complete the questionnaires.

### 2.4. Data Analysis

Initial data analysis was conducted in IBM SPSS Statistics, version 28. Missing data was observed for 6 of the cases (3.3%), all on the measure of perceived physical competence. Little MCAR’s test indicated that data was missing completely at random (χ^2^ (4) = 0.25, *p* = 0.99). The default full information maximum likelihood (FIML) procedure of Mplus [60] was used to account for missing data. FIML computes a likelihood function for each participant based on available data and is a highly recommended approach for handling missing values [61].

Next, two multilevel path models were constructed in Mplus, using the MLR estimator. In these models, class was included as a clustering variable to control for the nested structure of the data. Model fit was evaluated using the Chi square, root mean square error of approximation (RMSEA) and comparative fit index (CFI), using cut-off values of *p* < 0.05, 0.06, and 0.90, respectively [62]. In the first model, perceived physical competence, the frequency of sports participation and the type of sports were included as predictors of social acceptance in PE and social acceptance in the classroom. Gender and age were included in this model as well and related to all predictor and outcome variables. In addition, the frequency of sports participation was included as a predictor of perceived physical competence and a covariance between the frequency of sports participation and the type of sports was added. The second model was similar to the first one, but then grouped by the type of sports, meaning that separate models were estimated for children participating in individual sports and children participating in team sports.

## 3. Results

### 3.1. Descriptive Statistics

An overview of the descriptive statistics of the included variables is presented in Table 1.

### 3.2. Overall Relations

The first model, in which perceived physical competence, frequency of sports and type of sports were included as predictors of social acceptance in the classroom and in PE had an acceptable fit (χ^2^ (3) = 7.31, *p* = 0.06; RMSEA = 0.00; CFI = 1.00). Significant relations in this model are presented in Figure 1 (see Appendix A for the full model including non-significant relations). In total, a significant 6.6% of the variance of social acceptance in PE was explained by the model (*p* = 0.025). The amount of explained variance of social acceptance in the regular classroom was not significant (R^2^ = 0.02, *p* = 0.39). Frequency of sports participation was a significant predictor of social acceptance in PE (β = 0.22 (0.07), *p* < 0.001, 95%-CI = 0.09 to 0.35. Perceived physical competence (β = 0.13 (0.09), *p* = 0.15, 95%-CI = −0.05 to 0.31) and sport type (β = −0.09 (0.11), *p* = 0.38, 95%-CI = −0.30 to 0.11) were not significantly related to social acceptance in PE. Perceived physical competence (β = 0.03 (0.07), *p* = 0.67, 95%-CI = −0.11 to 0.17), sports frequency (β = 0.13 (0.08), *p* = 0.10, 95%-CI = −0.03 to 0.18) and type of sports (β = −0.06 (0.13), *p* = 0.62, 95%-CI = −0.31 to 0.18) were not significantly related to social acceptance in the regular classroom.

### 3.3. Relations Separated by Type of Sports

The second model, in which relations between perceived physical competence, sport frequency and social acceptance were estimated separately for children participating in team sports and individual sports, fitted the data well (χ^2^ (2) = 2.25, *p* = 0.33; RMSEA = 0.04; CFI = 1.00), see Appendix A (team sports) and Appendix A (individual sports) for the full model.

Significant relations in the model for children participating in team sports are presented in Figure 2. Perceived physical competence was not a significant predictor of social acceptance in PE (β = −0.04 (0.08), *p* = 0.65, 95%-CI = −0.19 to 0.12), nor in the regular classroom (β = −0.02 (0.10), *p* = 0.81, 95%-CI = −0.22 to 0.17). For children participating in team sports, sport frequency was a significant and positive predictor of social acceptance in PE (β = 0.20 (0.06), *p* < 0.001, 95%-CI = −0.09 to 0.32). That is, children who participated more in sports on average were also more socially accepted in PE than their peers who were less often involved in sports. In contrast, sport frequency was not a significant predictor of social acceptance in the regular classroom (β = 0.10 (0.06), *p* = 0.09, 95%-CI = −0.02 to 0.22).

Significant relations in the model for children participating in individual sports are presented in Figure 3. In the model for children participating in individual sports, perceived physical competence was a significant and positive predictor of social acceptance in PE (β = 0.29 (0.06), *p* < 0.001, 95%-CI = 0.17 to 0.42). Children participating in individual sports who perceived themselves as more competent in physical skills were also more socially accepted in PE. Perceived physical competence was not a significant predictor of social acceptance in the regular classroom (β = 0.08 (0.05), *p* = 0.12, 95%-CI = −0.17 to 0.14). Sport frequency was neither a significant predictor of social acceptance in PE (β = 0.11 (0.08), *p* = 0.15, 95%-CI = −0.04 to 0.27) nor in the regular classroom (β = 0.02 (0.09), *p* = 0.79, 95%-CI = −0.16 to 0.20).

## 4. Discussion

The main purpose of this study was to investigate the relations between perceived physical competence, sports participation (frequency and type) and social acceptance within primary school PE, compared to the regular classroom. In addition, we examined whether these relations differed depending on the type of sport children participated in. Outcomes indicated that only the frequency of sports participation was positively related to social acceptance in primary school PE. In addition, for children participating in team sports, the frequency with which they participated in sports was positively related to social acceptance in PE, whereas for children participating in individual sports, their perceived physical competence was related to their social acceptance. No significant relations were found in the regular classroom, neither overall, nor for children participating in individual vs. team sports.

### 4.1. Perceived Physical Competence

Unexpectantly, perceived physical competence was not related to children’s social acceptance, neither in the regular classroom, nor in PE. We expected perceived physical competence to be related to social acceptance, specifically in PE, through the behaviour that accompanies children’s perception of their own physical competence (e.g., putting in more effort when perceiving oneself as more competent) [23]. Yet, although perceived competence has been linked to effortful behaviour during primary and secondary school PE in previous studies [23,30,31,32], we did not directly examine children’s behaviour here, meaning that children who perceived themselves to be more physically competent do not necessarily need to have shown more effortful behaviour during PE. Additionally, children might not have based their valuing of peers on engagement during PE, but also on other behaviours, for example collaborative actions or prosocial behaviours [63]. For future studies, it would be interesting to examine whether specific behaviours that children engage in (e.g., effort expenditure, prosocial behaviour [63]) are predictive of their social acceptance in PE. As children are expected to base their liking of peers on the chances of success a peer brings in performance settings [64], it is likely that children largely base their decisions on with whom they like to collaborate on specific behaviours that are of importance for successful performance during PE [10].

### 4.2. Sports Participation

As expected, children’s frequency of sports participation was related to their social acceptance in PE. Children in primary school highly value sports [18,33,34,35], making frequent participation in sports a means for becoming socially accepted amongst peers [37]. Moreover, sports provide children with opportunities to build social and physical skills, both important factors for gaining social acceptance [5,18,21,36,45,46,47,48]. Our study extends prior research in which sports participation was found to be related to general social acceptance amongst adolescents [38,39] by showing that sports participation is an important predicting factor for social acceptance particularly in the PE context.

Unexpectedly, sports participation was not related to social acceptance in the classroom context. Although we already anticipated that relations between sports participation and social acceptance would be weaker in the classroom context compared to the PE context, we did expect sports participation to be related to acceptance in the regular classroom as well, given the high value that primary school children attach to sports [18,33,34,35]. Considering the assumed context specificity of social acceptance [10,14,15], this result is less surprising. As we asked children to indicate whether they would like to collaborate with their peers in PE or the regular classroom specifically, it is likely that children took characteristics or behaviours into account that determine good performance in the specific context involved. In this sense, sports participation will likely determine a child’s success in PE (i.e., a child that practices sports will be more successful than a child that is not involved in sports), but not necessarily in successfully completing assignments in the regular classroom, which is more dependent on children’s academic capabilities. In line with this idea, academic achievement has been found to be an important predictor of children’s social acceptance within the school context, but not in sports settings [65]. To gain more insight into the correlates of social acceptance, it would be worthwhile for future studies to simultaneously examine factors related to good performance in PE and the regular classroom.

Although we hypothesised that participating in team sports would be related to higher social acceptance, the type of sports children engaged in was not related to their social acceptance (in neither contexts). Our categorization of sports type may play a role in explaining this outcome. Specifically, the distinction between individual and team sports is less pronounced in childhood than in adolescence and adulthood [50,51]. Even in individual sports, children often train together in groups, making it more problematic to classify sports as purely individual sports in this age group. Further, children might not necessarily value the broader categories of individual and team sports differently but might rather value specific sports within these categories differently (e.g., football compared to basketball). Although the categorization of team sports and individual sports is commonly used in research [49], a more fine-grained categorisation might provide more insight into how sports participation relates to children’s social acceptance.

### 4.3. Individual vs. Team Sport Players

In addition, we examined whether the relation between frequency of sports participation and social acceptance differed depending on the type of sports primary school children participated in. In line with our expectations, frequency of participation in team sports was related to social acceptance in PE. This relation was not found for children participating in individual sports. The inherently social character of team sports is likely to provide team sporters with more opportunities to develop social skills and build social relationships [49,50,51], even more so when participating in their sports more frequently, contributing to their social acceptance [52]. Additionally, team sporters might play team sports together with their classmates, enabling them to show their physical competence to their peers in an additional setting outside of school (e.g., when playing football on the same team, children can see whom of their peers are successful football players and whom are not). Similar opportunities are not, or to a lesser extent, available for children participating in individual sports because they are less likely to participate in their sport together with classmates. As actual physical competence predicts social acceptance [10,22], the overt demonstration of physical competence in multiple settings for team sporters may explain why frequency of participation was related to social acceptance for team sporters but not for individual sporters.

Although we did not have specific hypotheses for individual sporters, perceived physical competence was found to be related to their social acceptance in PE. As argued previously, peers are more likely to base their opinion about a child on behaviours they perceive during PE (e.g., dominance) than on the child’s actual physical competence level, and children’s behaviour during PE is partly determined by their own perceived physical competence [30,31,32] This could also explain why perceived physical competence is related to social acceptance [23]. As individual sporters have fewer opportunities to show their skills to their peers in settings outside of school, the relation between perceived physical competence and social acceptance might apply more for individual sporters than team sporters.

The finding that children’s perceived physical competence adds to their social acceptance (specifically for individual sports participants) can have devastating consequences for children with low levels of perceived physical competence. Following our results, these children seem to be less socially accepted in PE, which may result in a negative vicarious circle: their lower perceived competence levels limit their chances for experiencing successes during PE, both physically and socially, negatively contributing to their perceived physical competencies and social acceptance even more [36]. It, thus, seems vital to ensure that children who perceive themselves as less physically competent receive the chance to experience successes during PE, in order for them to build feelings of competence and to gain social acceptance [23].

No relations were found between perceived physical competence, frequency of sport participation and social acceptance in the regular classroom, again underlining the context specificity of social acceptance [10,14,15], suggesting that sports participation might be less important for gaining social acceptance in the regular classroom.

### 4.4. Implications

PE is argued to be a context contributing to social acceptance [66,67]. Our results suggest that different factors play a role in social acceptance in primary school PE and the regular classroom, underlining the uniqueness of PE, and emphasising that PE teachers should be aware of how their lessons may impact children’s social development. Especially, children with lower perceived physical competence and/or less sports participation seem to be at risk (note: perceived physical competency and sport participation are related, so these may be the same children [24]). Not only physically, but also socially, the lower (perceived) physical competence and less frequent sports participation of these children may put them at risk of being less socially accepted in the PE classroom. This is a worrying finding, as social acceptance in PE is a major determinant of children’s experiences during PE, being predictive of—amongst others—PE enjoyment and effort expenditure, and motivation for sports and physical activity in general [66,68]. These are in turn important sources of primary school children’s physical activity behaviours [69], thereby, being of vital importance for children’s development, general health and wellbeing [54]. In preventing negative experiences during PE, PE teachers play a facilitating role by designing their classroom in such a way that they can foster positive social experiences for all students. They can for example use strategies such as grouping of students, offering options and choice, using a warm, caring and positive communication style and creating optimal challenge by providing exercises at different difficulty levels [68,70].

### 4.5. Limitations

Although this study yielded important outcomes, it also suffered from several limitations. Firstly, no conclusions can be drawn regarding causality. Although it seems reasonable to expect perceived physical competence and sports participation to be determinants of social acceptance [19], these relations might also work the other way around. It has been shown, for example, that children who are more socially accepted by their peers are more motivated for and have more opportunities to engage in physical activities [36,68], which may, consequently, affect their perceived physical competence [68,71]. Experimental or longitudinal studies are needed to disentangle the order of these effects.

Secondly, in assessing sports participation, children were only asked to indicate if, and if so, how often, they engaged in organised sports. However, children also engage in sports outside the context of organised sports clubs (e.g., playing in the park). This type of physical activity might be relevant to include in future studies as well [48].

Thirdly, we examined children’s frequency of participating in sports by asking about the number of days children were involved in their primary sports. Although this is considered a reliable and valid measure of sport participation [56], it provides a rather global indication and does not provide insight into whether children participate in multiple (types of) sports. For future research it is advised to use a more fine-grained measure, asking about all sports children are engaged in.

Lastly, differences between the PE teachers of participating classes may partly explain our results. That is, teachers play a key role in shaping a need-satisfying environment, in which children feel accepted and safe [72]. For example, more experienced teachers and teachers with a specialist training seem to be better able to adapt the PE classroom to their students’ needs [73,74]. It can be hypothesised that (predicting factors of) social acceptance in classes of these teachers will differ from classes where the environment is experienced as less accepting and need-supportive. We expect that this limitation will have played a minor role in our study, since all teachers had the same educational background (i.e., specialist teachers). Still, for future studies it seems of interest to examine to what extent teacher characteristics are predictive of children’s experiences of social acceptance in PE.

## 5. Conclusions

To conclude, the present study shows that the relations between perceived physical competence and social acceptance and between sports participation and social acceptance depend on the primary school context involved. That is, the frequency of sports participation was positively related to social acceptance in PE, whereas this relation was not found in the regular classroom. Moreover, when comparing children engaged in different types of sports, the frequency of sports was related to social acceptance for team sporters, but not for individual sporters. In contrast, perceived physical competence was related to social acceptance (although only in PE) for individual sporters, but not for team sporters. These relations were, again, only found in the PE context, underlining the uniqueness of PE for children’s social development. For future research into the correlates of social acceptance, it seems important to include specific primary school contexts and different types of sports. In designing PE classrooms, children’s physical competence and sports participation seem essential factors to take into account in order to provide all children with positive social experiences.

## Figures and Tables

**Figure 1 children-10-00568-f001:**
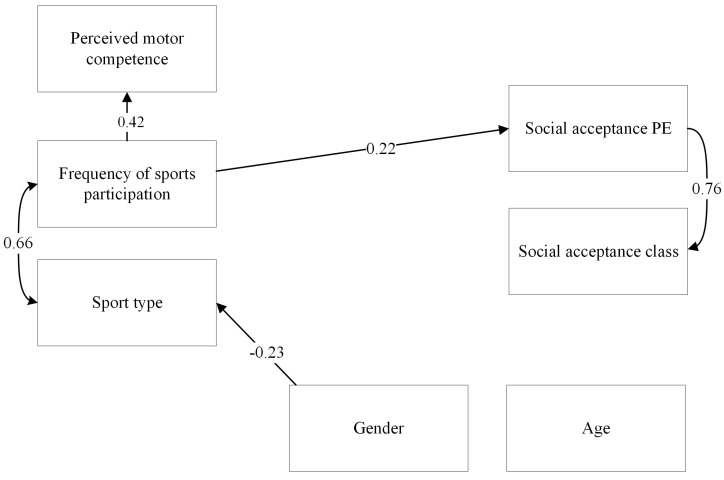
SEM model presenting significant paths between independent and dependent variables. Standardised path coefficients (betas) are presented in the figure.

**Figure 2 children-10-00568-f002:**
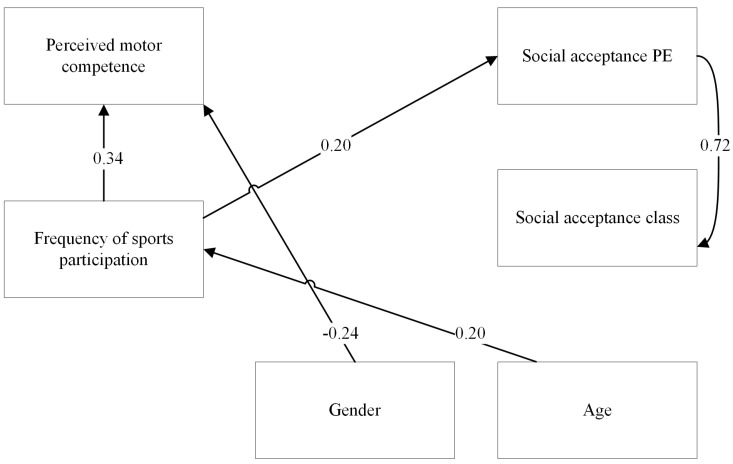
SEM model presenting significant paths between independent and dependent variables for children participating in team sports. Standardised path coefficients (betas) are presented in the figure.

**Figure 3 children-10-00568-f003:**
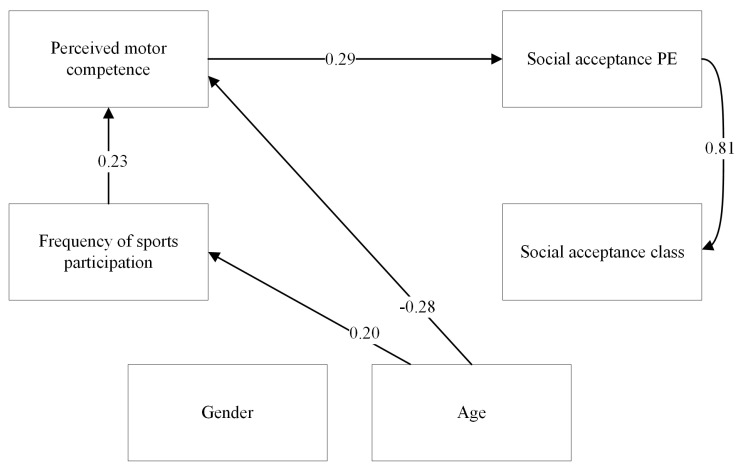
SEM model presenting significant paths between independent and dependent variables for children participating in individual sports. Standardised path coefficients (betas) are presented in the figure.

**Table 1 children-10-00568-t001:** Mean scores on sports frequency, type of sports, perceived motor competence and social acceptance in PE and the regular classroom.

	Total		Individual Sports (*n* = 66)		Team Sports (*n* = 94)	
	Mean (SD)	Min.–Max.	Mean (SD)	Min.–Max.	Mean (SD)	Min.–Max.
Perceived motor competence	3.11 (0.45)	1.96–4.00	2.99 (0.46)	2.04–4.00	3.24 (0.42)	1.96–4.00
Sport frequency	2.27 (1.29)	0–7	1.83 (1.17)	0–7	3.01 (0.79)	1–5
Social preference PE	0.00 (1.55)	−4.18–3.13	−0.05 (1.62)	−4.18–2.95	0.11 (1.52)	−3.54–3.13
Social preference classroom	0.00 (1.58)	−4.42–3.34	0.08 (1.73)	−4.42–3.34	−0.01 (1.50)	−3.70–3.01

## Data Availability

Data is available from the first researcher upon reasonable request.

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
