# Peer review of "Social Acceptance in Physical Education and the Regular Classroom: Perceived Motor Competency and Frequency and Type of Sports Participation"

_children, 2023, doi:10.3390/children10030568_

Round 1
Reviewer 1 Report
Congratulations to the authors for the article. The approach of the article is very interesting and methodologically quite good. It is well written and the discussion is well founded. It is only recommended to consider including "type" of sport in the title. Since by putting only participation in sport, it can confuse about the participation or not in it. Something that has not been considered, as all the participants are either individual or collective sport practitioners.
Author Response
Dear reviewer,
Thank you for your kind words about our work.
We agree with your comment and changed the title into "Social acceptance in physical education and the regular classroom: perceived motor competency and frequency and type of sports participation" - this way also including the type of sports participation.
Reviewer 2 Report
Thank you for allowing me to review this interesting study. Overall, the study has raised a very interesting point of discussion. I believe that this study has provided novel findings in this area, allowing readers to think more deeply about what is happening around motor or motor engagement in Physical Education classes, even more so with the boom that this activity has been having in recent times. discipline and its correct methodology.
First of all, I would like to share the need to carry out works like the one you present. They are necessary for the advancement of science in the field they study. The objective of the manuscript is clear and consistent. The study has been an interesting reading, it is necessary to know the reality of the sector on which the work emphasizes.
The abstract includes the necessary elements: background with purpose (objective) of the study, methods, results, main conclusions without exaggerating them.
In the introduction, sufficient ordered references of the publications considered key, with significant and sufficient evidence, are indicated. I find the review of the literature really interesting, although some recent work should be added, such as:
Likewise, reasons are highlighted that justify the importance in a broad context and the current state of the subject investigated. The study is clearly defined and indicates the intention and meaning of the work. The objective to be tested in the study is recorded. The text is understandable and makes clear the main objective of the work and the main conclusions.
In relation to the material and methods, say that the study is described in detail. In addition to the methods, the intervention requirements are indicated in sufficient detail.
In general, it is a very interesting manuscript, despite some questions that are suggested to improve the manuscript and the study findings. It is recommended to add motor or motor competence as a keyword.
Don't you think that the training of the teachers of the students to whom the research has been carried out may have influenced the results? I would like this appreciation to be included in limitations of the work.
Author Response
We would like to thank the reviewer for the kind words.
In response to the suggestions:
- The reviewer mentions recent work that should be added, but without references. We would like to hear a follow-up on this remark, so we can take these suggestions into account.
- We have added motor competence as a keyword.
- Following the reviewer's suggestion, differences in experience and training of the students' teacher might indeed have been a limiting factor of the study. We have added this limitation to the manuscript.